# MLLM CAN SEE? DYNAMIC CORRECTION DECODING FOR HALLUCINATION MITIGATION

**Chenxi Wang**[♠]*, **Xiang Chen**[♡]*, **Ningyu Zhang**[♠]†, **Bozhong Tian**[♠], **Haoming Xu**[♠],
**Shumin Deng**[♢], **Huajun Chen**[♠♣]†

[♠]Zhejiang University   [♡]Nanjing University of Aeronautics and Astronautics
[♢]National University of Singapore, NUS-NCS Joint Lab, Singapore
[♣]Zhejiang Key Laboratory of Big Data Intelligent Computing
{sunnywcx,zhangningyu}@zju.edu.cn, xiang_chen@nuaa.edu.cn

## ABSTRACT

Multimodal Large Language Models (MLLMs) frequently exhibit hallucination phenomena, but the underlying reasons remain poorly understood. In this paper, we present an empirical analysis and find that, although MLLMs incorrectly generate the objects in the final output, they are actually able to recognize visual objects in the preceding layers. We speculate that this may be due to the strong knowledge priors of the language model suppressing the visual information, leading to hallucinations. Motivated by this, we propose a novel dynamic correction decoding method for MLLMs (**DeCo**), which adaptively selects the appropriate preceding layers and proportionally integrates knowledge into the final layer to adjust the output logits. Note that DeCo is model agnostic and can be seamlessly incorporated with various classic decoding strategies and applied to different MLLMs. We evaluate DeCo on widely-used benchmarks, demonstrating that it can reduce hallucination rates by a large margin compared to baselines, highlighting its potential to mitigate hallucinations[1].

*"The first principle is that you must not fool yourself—and you are the easiest person to fool."*

— Richard Feynman

## 1 INTRODUCTION

Recently, the rapid development of Multimodal Large Language Models (MLLMs) has demonstrated a potential pathway towards achieving Artificial General Intelligence (AGI) (Wang et al., 2024b; Yao et al., 2024; Lu et al., 2024a; Team, 2024; OpenAI, 2023; Liu et al., 2023b; Chern et al., 2024). However, in practice, the development of MLLMs is hindered by the phenomenon of hallucination, which typically results in the model generating statements about non-existent images while neglecting to mention certain visible objects, effectively causing it to fool itself (Bai et al., 2024; Liu et al., 2024a; Li et al., 2023b; Liu et al., 2023a; Rawte et al., 2023). This issue poses significant risks in high-stakes fields such as medical imaging (Chen et al., 2024b; Hu et al., 2023; Wang et al., 2023c), autonomous driving (Cui et al., 2024; Wang et al., 2023d), and human-computer interaction systems (Brie et al., 2023), where such errors could result in irreparable consequences.

The reasons behind hallucinations in MLLMs are complex. Unlike analyses focused on unimodal LLMs (Chuang et al., 2024; Chen et al., 2024d; Orgad et al., 2024; Chen et al., 2024e; Lu et al., 2024b; Wang et al., 2024a), many current works assume that MLLM may indeed 'see' visual information. However, due to factors such as excessive model depth (Chen et al., 2024c; Zhang et al., 2024a), aggregation patterns (Huang et al., 2024), or priors knowledge inherent in the MLLMs (Leng et al., 2023; Zhang et al., 2024b), these models ultimately still experience hallucinations (The details can be found in Appendix B). Concretely, our understanding of the underlying mechanisms of hallucinations in MLLMs remains limited. It is still uncertain whether the visual information is never correctly recognized or if it is recognized but subsequently suppressed by later information streams.

---

* Equal Contribution.
† Corresponding Author.
[1]Code is available at https://github.com/zjunlp/DeCo.

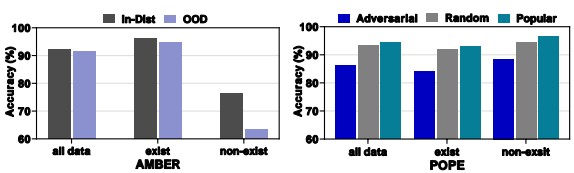 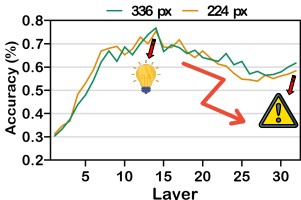

(a) Object probing results.  (b) Different resolution results.

Figure 1: Overall results of the probing experiment with MLLMs, indicating that they possess a certain level of awareness regarding the presence of visual objects (Figure 1(a)), with prediction accuracy being higher in the preceding layers (Figure 1(b)) but gradually **decline** afterward.

**Hallucinated MLLM can see (to some extent).** Inspired by the aforementioned works, we conduct an empirical analysis and find that MLLMs are not blind; they can recognize objects in the preceding layers, but this recognition is suppressed in later layers, leading to hallucinations. Specifically, we focus on object hallucinations[2] and conduct experiments with MLLMs, demonstrating that they know to some extent whether an object exists (as shown in Figure 1 and Section 2.1). We further observe that the confidence of generated tokens is influenced by the knowledge priors of MLLMs (Section 2.2), leading to a reduction in the probability of ground truth tokens in the deeper layers.

**Dynamic correction decoding with preceding-layer knowledge.** Based on those findings, we propose **D**ynamic Correction **D**ecoding with pre**C**eding-Layer Kn**o**wledge (**DeCo**) to mitigate hallucinations for MLLMs. Our core hypothesis is that preceding layers exhibit higher confidence for ground truth tokens, and the logits for these tokens should rank prominently at the last layer's outputs. To enhance the logits of ground truth tokens, DeCo dynamically selects preceding layer and utilizes its prior knowledge to correct the final output logits. Additionally, we introduce a dynamic soft modulation to preserve the original style of the generated responses. DeCo is training-free and can be integrated with any popular decoding strategies, such as greedy search, nucleus sampling as well as beam search, and can seamlessly incorporate into any MLLMs for hallucination mitigation.

**Contributions.** Our primary contribution lies in exploring the internal mechanisms of hallucinations in MLLMs. We find that the confidence of generated tokens is influenced by the knowledge priors of MLLMs, leading to a reduction in the probability of ground truth tokens in the deeper layers. We further propose DeCo, a dynamic correction decoding method guided by preceding-layer knowledge. DeCo is integrated with InstructBLIP, MiniGPT-4, LLaVA, and Qwen-VL using three popular decoding strategies: greedy search, nucleus sampling, and beam search. Experimental results show that DeCo achieves an average hallucination suppression rate of **10.8%** in image captioning dataset, demonstrating superior suppression effectiveness. Additionally, DeCo outperforms baselines on visual question answering datasets including POPE, and MME. Additionally, we analyze the latency and throughput, showing that DeCo introduces an approximate 1.2x increase in latency compared to the basic decoding process, much faster than previous baselines such as VCD and OPERA.

## 2 WHY DO MLLMs GENERATE NON-EXIST OBJECTS?

In this section, we conduct a series of empirical analysis to investigate the internal mechanisms of MLLM and elucidate the underlying reasons for its generation of non-existent objects. To strike a balance between the realism and complexity of the experiments, we primarily focus on the generation of objects in image description scenarios (image caption tasks).

**Preliminaries of MLLM generation.**    MLLMs typically concatenate visual tokens, processed by the visual encoder and projection layer, with embedded textual tokens before feeding them into an autoregressive language model. We denote the visual tokens as $\mathbf{X}^V = \{x_{v_1}, x_{v_2}, \ldots, x_{v_P}\}$ and textual tokens as $\mathbf{X}^C = \{x_{c_1}, x_{c_2}, \ldots, x_{c_Q}\}$. Here $P$ and $Q$ are the lengths of the visual tokens

---

[2]This approach is applicable to other types of hallucinations as well.

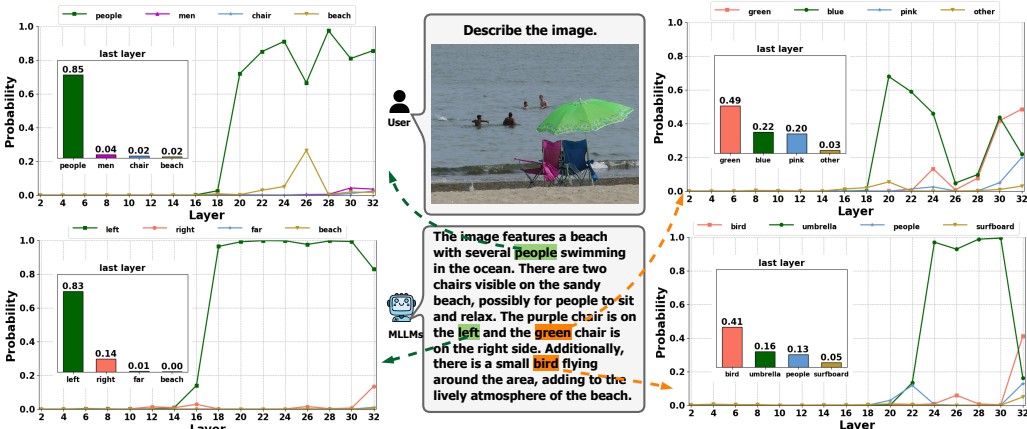

Figure 2: Illustration of token probabilities across transformer layers, which reveals distinct trends for target hallucinated (**orange**) and non-hallucinated (**green**) tokens. In the preceding layers, non-hallucinated tokens exhibit a higher probability. In the final layers, hallucinated tokens demonstrate **increased probabilities**, while the probability of non-hallucinated tokens **drops sharply**.

and textual tokens respectively. Finally, the input is $\mathbf{X} = \text{concat}\{\mathbf{X}^V, \mathbf{X}^C\}$. Then $\mathbf{X}$ would be passed into MLLM with $\mathbf{N}$ stacked transformer layer. The intermediate variable generated by the $i$-th layer is called hidden states, denoted as $\mathbf{h}^i = \{h_0^i, h_1^i, \ldots, h_{T-1}^i\}$, where $T = P + Q$. During the generation phase, we use the hidden state at the last position in the final layer, which is mapped to the vocabulary dimension through an affine layer $\phi(\cdot)$, to predict the probability of the next token. Formally, we have:

$$p(x_T | x_{<T}) = \text{softmax}(\phi(h_{T-1}^N))_{x_T}, x_T \in \mathcal{V} \tag{1}$$

where we use $x_{<T}$ to simplify the sequence $\{x_i\}_{i=0}^{T-1}$ and $\mathcal{V}$ refers to the whole vocabulary set.

## 2.1 FINDING 1: MLLM KNOWS TO SOME EXTENT WHETHER AN OBJECT EXISTS

Inspired by (Ye et al., 2024), we explore how MLLMs comprehend objects in the image captioning task. For simplicity, we abstract this process into a function called **isexist(obj)**, which determines whether an object is present in an image. To examine the application of this function within the MLLM's image captioning workflow, we conduct probing experiments at the conclusion of object descriptions in each layer of the MLLM's language model component, which consists of 32 transformer layers in a 7-billion-parameter model (Detailed setup in Appendix C.1).

We employ the prompt template, *"USER: <image>Describe the image. ASSISTANT: The image contains obj."* Both the training and testing datasets are formatted accordingly before being input into MLLMs. We train a probe classifier at the final position of the hidden state outputs for each transformer layer, resulting in a total of 32 classifiers. (For details on the subset division, OOD and in-distribution splits, and prompt templates, please refer to Appendix C.1.) The model is evaluated using the test set, as shown in Figure 1(a) (left). Further experiments are conducted on three splits of the evaluation dataset proposed by POPE, with results reported in Figure 1(a) (right). These evaluations provide a comprehensive understanding of the model's object recognition capabilities across diverse scenarios.

We select the best-performing probe classifier from the 32 classifiers to compare accuracy across all objects, existing objects, and non-existing objects. Our results show that the MLLM achieves high accuracy for correctly generated objects in image captions. Despite generating many non-existent objects, the MLLM still maintains around 80% accuracy in our probing experiments. This suggests that **MLLMs possess a certain level of understanding regarding object existence in images**.

Additionally, our probing experiments reveal higher accuracy in the preceding layers, as illustrated in Figure 1(b), which aligns with previous findings (Zhang et al., 2024b; Leng et al., 2023). Furthermore, we show that increasing the resolution of the visual encoder (from 224px to 336px) enhances

accuracy for non-existing objects, indicating that **token information at the last position in the preceding layers better represents visual information**. (For a detailed explanation of the different visual resolutions, please refer to Appendix C.1). These findings suggests that the utilization of the preceding-layers in MLLMs enables the model to perform self-correction.

## 2.2 FINDING 2: LANGUAGE MODEL PRIORS SUPPRESS THE VISUAL INFORMATION THAT MLLM ALREADY SEE.

We hypothesize that the representations in the preceding layers effectively capture (to some extent) visual information. However, the prior knowledge embedded in the MLLM reduces the probabilities of ground truth tokens in deeper layers. Figure 2 illustrates this hypothesis with running examples. We analyze the Top-4 tokens ranked by probability in the final layer's output. Non-hallucinated tokens like *"people"*, *"left"*, *"blue"*, and *"umbrella"* exhibit high probabilities from the 18th layer. In contrast, hallucinated tokens like *"bird"* and *"green"* only show comparatively high probabilities around the 30-th layer. Interestingly, the probabilities of ground truth tokens *"umbrella"* and *"blue"* sharply decline from the 30-th layer onwards, eventually falling below the hallucinated tokens' probabilities in the final layer.

To further investigate this phenomenon, we conduct an early exit experiment (Teerapittayanon et al., 2016; Elbayad et al., 2020; Schuster et al., 2022) to analyze the evolution of the MLLM's internal representations across transformer layers. We randomly select 500 images from the MSOCO dataset and use random prompts to elicit raw responses from LLaVA-1.5-7b. We then extract all non-existent objects along with their corresponding preceding text and input this data into the MLLM. We observe the probabilities of the next token across the transformer layers to gain insights into the model's behavior (see Appendix C.2 for detailed experimental setup). The output of the $i$-th layer is denoted as $h^i$, and the probability

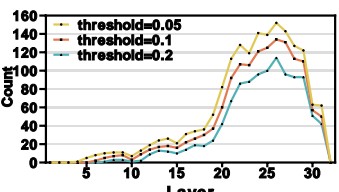

Figure 3: Distribution of activated ground-truth tokens across layers.

distribution of the next token is represented as $p(\cdot|x_{<s})^i = \mathrm{softmax}(\phi(h^i_{s-1}))$. To reduce the observation tokens and simulate the real sampling process, we truncate the vocabulary, similar to Top-$p$ sampling, and obtain the candidate tokens, denoted as $\mathcal{V}_{candidate}$ with a default threshold of 0.9. We then label the tokens in $\mathcal{V}_{candidate}$. Specifically, we filter out data where $\mathcal{V}_{candidate}$ contains at least one ground truth token and observe whether an **activated ground truth token** exists among the candidate tokens, formally expressed as:

$$\exists \, x_a \in \mathcal{V}_{candidate} \wedge i \in (0, N], \; p(x_a|x_{<s})^i - p(x_h|x_{<s})^i \geq \text{threshold}, \tag{2}$$

where $x_a$ is the activated ground truth token, $x_h$ is the token with the highest probability of being a hallucinated token in the probability distribution of the final layer and threshold $\in (0, 1)$. Based on the experimental setup described above, we conducted the following investigation:

**What suppresses the expression of visual facts?** We analyze the occurrence of $x_a$ at each decoding layer, as shown in Figure 3. The results reveal that the activated ground truth tokens are primarily present between layers 20 and 28, indicating that MLLMs accurately recognize the image content in the latter layers. Notably, differences in experimental setups account for the variation in interval layers observed between Finding 1 and Finding 2. However, the activated ground truth tokens are suppressed in the final output layer. This suppression may stem from the guidance of the input image or the inherent knowledge bias of the MLLM. To investigate this, we generate candidate tokens $\mathcal{V}'_{candidate}$ in the absence of an input image, representing tokens based on the MLLM's inherent knowledge. We calculate that the overlap rate of $x_h$ existing in $\mathcal{V}'_{candidate}$ reaches 91.05%, suggesting that even without expressing image information, MLLMs still tend to generate the original hallucination tokens. This finding reveals that **the inherent knowledge in MLLMs may diminish the probability of the ground truth token in the deeper layers.**

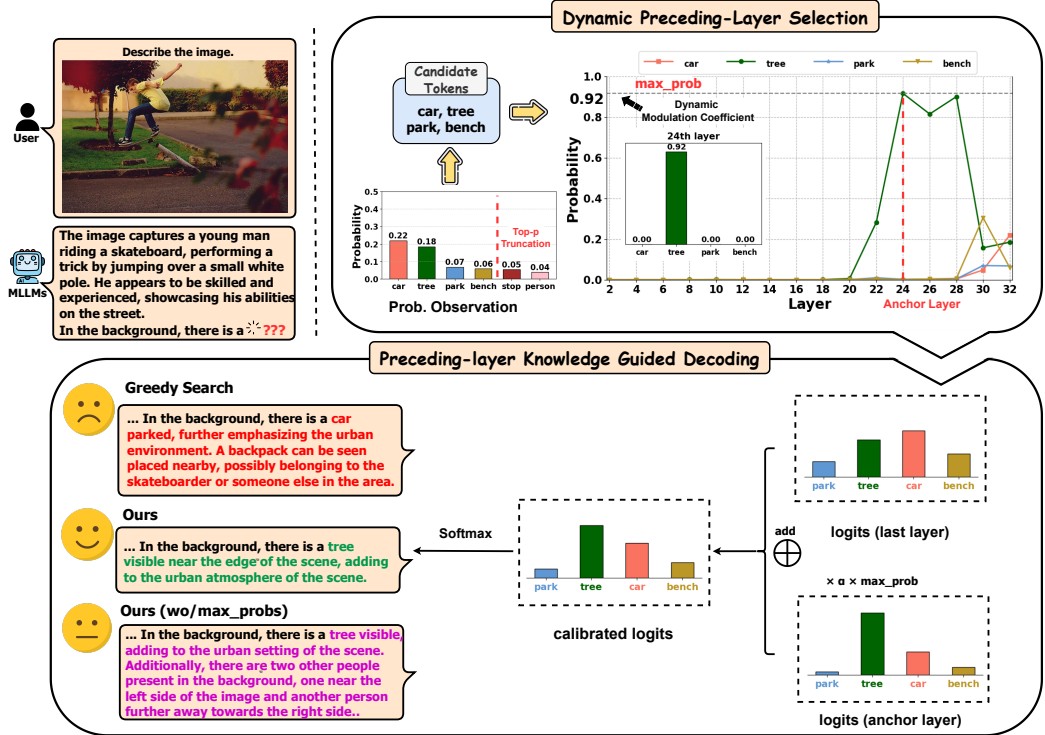

Figure 4: Framework of DeCo. DeCo first dynamically selects an appropriate anchor layer from the preceding layers and then correct the knowledge in the final layer with dynamic coefficient.

# 3 PROPOSED APPROACH: DYNAMIC CORRECTION DECODING WITH PRECEDING-LAYER KNOWLEDGE

After investigating the reasons why MLLMs generate non-existent objects, inspired by (Chuang et al., 2024), we introduce **D**ynamic Correction De**c**oding with pre**C**eding-Layer Kn**o**wledge (**DeCo**), which can alleviate hallucinations during inference. The overall framework of Deco is illustrated in Figure 4, consisting of dynamic preceding layer selection (Section 3.1) and decoding correction with preceding-layer knowledge (Section 3.2).

## 3.1 DYNAMIC PRECEDING-LAYER SELECTION

**Candidate token acquisition.** Due to the vast vocabulary space, we track only the changes in the top-ranked tokens as candidate tokens across different layers for computational convenience. This is based on the hypothesis that ground tokens usually appear in the top position of the MLLM's last layer output logits. Inspired by (Li et al., 2023a), we use a truncation strategy to select the candidate tokens, with the default truncation strategy being top-$p$ truncation, formally:

$$\mathcal{V}_{\text{candidate}}\left(x_T | x_{<T}\right) = \left\{ x_T \in \mathcal{V} : \sum_{v \in \mathcal{V}p} P_\tau(x_T = v | x_0, x_1, \ldots, x_{T-1}) \leq p \right\} \quad (3)$$

where $\mathcal{V}$ is the whole vocabulary, and $p$ refers to the parameter used in top-$p$. The selected candidate tokens are theoretically ensured to be of high quality, thereby preventing the inclusion of low quality tokens (e.g.,semantically incorrect tokens) that exhibit high probabilities in preceding-layers but low probabilities in the final layer.

Table 1: Hit Rate of layers across different intervals.

| Layer Range | 20-28 | 15-28 |
|---|---|---|
| Hit Rate (%) | 61.69 | 71.14 |

**Preceding-layer selection.** Our findings in Section 2 demonstrate that activated ground truth tokens typically exhibit higher probabilities in preceding layers compared to hallucinated tokens. Based on this observation, we hypothesize that selecting the token $x_{th}$, where $x_{th} \in \mathcal{V}_{candidate}$, with the highest probability from the interval layers corresponds to the ground truth token. We compute the accuracy of $x_{th}$ as the ground truth token and denote this metric as the hit rate, as shown in Table 1. The results indicate that within a specific range of layers (e.g., 15-28), $x_{th}$ indeed has a high universal probability of representing the ground truth token. Intuitively, we track candidate tokens and dynamically choose the layer in which the token with the highest probability among the preceding layers resides to calibrate the final logit distribution of the MLLM. The selected preceding layer is referred to as the anchor layer, formally defined as:

$$\mathcal{A} = \text{argmax}_i \big\{ x_T \in \mathcal{V}_{\text{candidate}} : \text{softmax}(\phi(h^i_{T-1}))_{x_T}, i \in [a, b] \big\}, \tag{4}$$

where $a \leq b, a, b \in [1, N]$, and $[a, b]$ represents the layer interval for MLLMs. Expanding the range of layers can improve the hit rate. To avoid increased search computation time, we assign default values of $a = 20$ and $b = 28$ for our subsequent experiments.

## 3.2 DECODING CORRECTION WITH PRECEDING-LAYER KNOWLEDGE

**Dynamic soft modulation.** We introduce a dynamic modulation coefficient, defaulting to the maximum probability. Formally, we have:

$$\text{max\_prob} = \max(\text{softmax}(\phi(h^{\mathcal{A}}_{T-1}))). \tag{5}$$

This coefficient can help prevent hard changes in logits, particularly when the probability differences between candidate tokens in preceding layers are insignificant. From the example in Figure 4, we can observe that the absence of the dynamic modulation coefficient may lead to semantic incoherence or even more severe hallucinations.

**Preceding-layer knowledge guided decoding.** Given the selected preceding layers, we integrate information from these layers into the final layer to correct the logit distribution. We utilze a hyperparameter, $\alpha$, to control the proportion of early-layer information incorporated. Additionally, dynamic soft modulation is employed to preserve the generative style of the original model. By utilizing the correction of preceding-layer representations, the probability of predicting the next token and the logits are updated as follows:

$$\hat{p}(x_T \mid x_{<T}) = \text{softmax}(\text{logits})_{x_T}, \tag{6}$$

$$\text{logits} = \phi(h^N_{T-1}) + \alpha \times \text{max\_prob} \times \phi(h^{\mathcal{A}}_{T-1}), \tag{7}$$

where $N$ is the last layer of MLLM and $\mathcal{A}$ is the selected preceding layer.

## 4 EXPERIMENT

### 4.1 SETUP

**Baselines.** We integrate DeCo with various decoding methods, including greedy decoding, nucleus sampling, and beam search, and compare it against several baselines for mitigating hallucinations, as outlined below: Dola (Chuang et al., 2024) is specifically designed for alleviating hallucinations in factual tasks for LLMs by reducing shallow semantic influences to improve the factuality of the final layer's output. VCD (Leng et al., 2023) mitigates the influence of language model's priors in MLLMs by generating representations that enhance visual information through the subtraction of interfering knowledge prior during each sampling step. OPERA (Huang et al., 2024) dynamically penalizes overconfident tokens based on the emergence of aggregation patterns, while proposing a retrospective allocation strategy to avoid cases where hallucinations have already occurred. For all the baselines, we use the default hyperparameters from the source code for a fair comparsion.

Table 2: **CHAIR hallucination evaluation results**. Lower scores indicate fewer hallucinations. OPERA utilizes beam search, VCD applies nucleus sampling, and DeCo is the proposed method compatible with various decoding approaches.

| Decoding | Method | InstructBLIP | | MiniGPT-4 | | LLaVA-1.5 | | Qwen-VL | |
| | | CHAIR$_S$ $\downarrow$ | CHAIR$_I$ $\downarrow$ | CHAIR$_S$ $\downarrow$ | CHAIR$_I$ $\downarrow$ | CHAIR$_S$ $\downarrow$ | CHAIR$_I$ $\downarrow$ | CHAIR$_S$ $\downarrow$ | CHAIR$_I$ $\downarrow$ |
|---|---|---|---|---|---|---|---|---|---|
| Greedy | Vanilla | 58.8 | 23.7 | 31.8 | 9.9 | 45.0 | 14.7 | 46.0 | 12.5 |
| | DoLa | 48.4 | 15.9 | 32.2 | 10.0 | 47.8 | 13.8 | 46.8 | 12.9 |
| | **DeCo (Ours)** | **41.2** $\downarrow$**17.6** | **14.4** $\downarrow$**9.3** | **27.0** $\downarrow$**4.8** | **8.8** $\downarrow$**1.1** | **37.8** $\downarrow$**7.2** | **11.1** $\downarrow$**3.6** | **42.2** $\downarrow$**3.8** | **10.7** $\downarrow$**1.8** |
| Beam Search | Vanilla | 55.6 | 15.8 | 30.6 | 9.5 | 48.8 | 13.9 | 41.8 | 10.8 |
| | OPERA | 46.4 | 14.2 | 26.2 | 9.5 | 44.6 | 12.8 | 34.6 | 9.5 |
| | **DeCo (Ours)** | **43.8** $\downarrow$**11.8** | **12.7** $\downarrow$**3.1** | **24.8** $\downarrow$**5.8** | **7.5** $\downarrow$**2.0** | **33.0** $\downarrow$**15.8** | **9.7** $\downarrow$**4.2** | **32.0** $\downarrow$**9.8** | **8.7** $\downarrow$**2.1** |
| Nucleus | Vanilla | 54.6 | 24.8 | 32.6 | 10.7 | 48.8 | 14.2 | 49.2 | 13.1 |
| | VCD | 58.0 | 17.0 | 33.8 | 11.1 | 54.0 | 16.0 | 46.4 | 11.9 |
| | **DeCo (Ours)** | **43.6** $\downarrow$**11.0** | **12.9** $\downarrow$**11.9** | **30.8** $\downarrow$**1.8** | **9.5** $\downarrow$**1.2** | **42.8** $\downarrow$**6.0** | **13.2** $\downarrow$**1.0** | **43.8** $\downarrow$**5.4** | **11.8** $\downarrow$**1.3** |

Table 3: **POPE hallucination evaluation results**. The best results are in bold.

| Decoding | Method | InstructBLIP F1 $\uparrow$ | MiniGPT-4 F1 $\uparrow$ | LLaVA-1.5 F1 $\uparrow$ | Qwen-VL F1 $\uparrow$ |
|---|---|---|---|---|---|
| Greedy | Vanilla | 80.0 | 58.5 | 82.2 | 85.2 |
| | DoLa | 83.4 | 72.8 | 83.2 | 85.8 |
| | **DeCo (Ours)** | **84.9** $\uparrow$**4.9** | **77.4** $\uparrow$**18.9** | **86.7** $\uparrow$**4.5** | **86.3** $\uparrow$**1.1** |
| Beam Search | Vanilla | 84.4 | 70.3 | 84.9 | 85.3 |
| | OPERA | 84.8 | 73.3 | 85.4 | 86.1 |
| | **DeCo (Ours)** | **84.9** $\uparrow$**0.5** | **77.9** $\uparrow$**7.6** | **86.7** $\uparrow$**1.8** | **86.4** $\uparrow$**1.1** |
| Nucleus | Vanilla | 79.8 | 52.8 | 83.1 | 84.5 |
| | VCD | 79.9 | 56.0 | 83.1 | 84.7 |
| | **DeCo (Ours)** | **81.8** $\uparrow$**2.0** | **63.8** $\uparrow$**11.0** | **85.4** $\uparrow$**2.3** | **85.2** $\uparrow$**0.7** |

**Model.** We select four of the most representative MLLM models for evaluation, including Instruct-BLIP (Dai et al., 2023), MiniGPT-4 (Zhu et al., 2024), LLaVA-1.5 (Liu et al., 2023b) and Qwen-VL (Bai et al., 2023). All the MLLMs we used have a language model size of 7 billion parameters (7B).

**Implementation Details.** To select the appropriate preceding layers for hallucination mitigation, we conduct ablation experiments, details of which can be found in the Section 4.4. For a 7B-sized, 32-layer decoder-only architecture language model, we choose layers 20-28 as candidates for the preceding layers (according to the findings in Section 2.1). For the image captioning and VQA tasks, $\alpha$ is set within the range of 0.1 to 0.6. In all experiments, we conduct inference on a single A800 GPU. The inference of 500 image-caption pairs take approximately 40 minutes.

## 4.2 BENCHMARK AND METRICS

**CHAIR.** Caption Hallucination Assessment with Image Relevance (CHAIR) (Rohrbach et al., 2018) metric, widely used in image captioning, identifies hallucinated objects by comparing the extracted objects with ground truth labels and evaluates both at the instance level (CHAIR$_I$) and sentence level (CHAIR$_S$), as shown in Eq. 8. Following (Huang et al., 2024), we conduct experiments using the same settings, including the consistent 500 images from the MSCOCO 2014 validation dataset and the identical prompt, "`Please help me describe the image in detail.`".

$$\text{CHAIR}_I = \frac{|\{\text{hallucinated objects}\}|}{\text{all mentioned objects}}, \text{CHAIR}_S = \frac{|\{\text{captions with hallucinated objects}\}|}{\text{all captions}}. \quad (8)$$

**POPE.** The Polling-based Object Probing Evaluation (POPE) (Li et al., 2023b) is a VQA-based metric for assessing object hallucination in MLLMs. It evaluates hallucinations by asking questions such as "`Is there a <object> in the image?`" where <object> is derived from three types of splits: random (randomly selected objects), popular (frequently occurring objects), and adversarial (objects closely related to those in the image). The evaluation includes 500 MSCOCO images, with six questions per image for each split. We use F1 score for performance evaluation.

**MME.** The comprehensive MLLM Evaluation benchmark (MME) (Fu et al., 2023) assesses the perceptual and cognitive abilities of MLLMs across a total of 14 subtasks, including tasks such as OCR, visual knowledge, attribute relationships, and object recognition.

**GPT-4o assisted evaluation.** To further assess the model's performance in image captioning, we extend beyond the CHAIR metric, which targets object hallucination. Following prior studies (Huang et al., 2024; Leng et al., 2023), an open evaluation is conducted using GPT-4o on 100 randomly sampled COCO images. GPT-4o assesses two assistants' descriptions in terms of Accuracy (A) (e.g., truthfulness), Detailedness (D) (e.g., richness) and Coherence (C). We introduce the prompt used in the experiments in Table 10 and Table 11.

## 4.3 EXPERIMENTAL RESULTS

**Results of hallucination in image captioning.** Note that we use the baseline's original decoding settings for a fair comparison and run DeCo under the same settings. From Table 2, we notice that DeCo consistently outperforms other approaches in mitigating hallucinations across four MLLMs—InstructBLIP, MiniGPT-4, LLaVA-1.5, and Qwen-VL—using three decoding strategies: greedy search, beam search, and nucleus sampling. We find that DeCo slightly outperforms OPERA, while our method demonstrates higher efficiency and simplicity in inference (see Section 4.4). Additionally, VCD does not perform as well, likely due to producing an increased number of hallucinated descriptions during the generation process. In conclusion, the proposed approach DeCo effectively reduces hallucinations in visual description tasks solely through dynamic decoding correction, achieving an average suppression rate of approximately **10.8%** on image captioning datasets. Additionally, we further evaluate the performance of DeCo on the AMBER image caption dataset, as detailed in Table 7 of the Appendix.

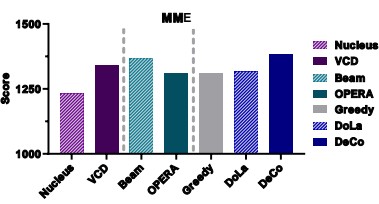

Figure 5: DeCo generally improves the MLLM's performance.

**Results of hallucination in VQA.** In contrast to image captioning, POPE employs a simple polling approach to assess hallucination levels in MLLMs with respect to object recognition. As shown in Table 3, DeCo demonstrates superior performance across all settings, further validating the effectiveness of the proposed approach. Additionally, Figure 5 reveals that DeCo also achieves better results on MME, which evaluates the multifaceted VQA capabilities of LLaVA-1.5. These findings suggest that the underlying mechanism we identified not only applies to object recognition but also extends to attribute-related tasks and more complex reasoning tasks.

**Results of GPT-4o's assistance.** Following (Huang et al., 2024; Leng et al., 2023), we further use GPT-4o to evaluate our method against greedy decoding across four distinct models. From Table 4, we notice that our approach consistently outperform greedy decoding in terms of accuracy, demonstrating its efficacy in hallucination suppression. The impact of decoding intervention is evident in the level of detail produced: for some models, our method yield only marginally higher or, in certain cases, slightly lower levels of detail compared to greedy decoding. DeCo also exhibits a coherence level comparable to that of the baseline. Nonetheless, our method exhibit a clear advantage in mitigating hallucinations across all evaluated models.

Table 4: GPT-4o assisted hallucination evaluation results on MSCOCO. Three aspects are verified, accuracy ($A$), detailedness ($D$) and coherence ($C$).

| Method | InstructBLIP | | | MiniGPT-4 | | | LLaVA-1.5 | | | Qwen-VL | | |
|---|---|---|---|---|---|---|---|---|---|---|---|---|
| | $A$ | $D$ | $C$ | $A$ | $D$ | $C$ | $A$ | $D$ | $C$ | $A$ | $D$ | $C$ |
| Greedy Search | 4.92 | 5.65 | 6.89 | 5.71 | **6.20** | **7.67** | 5.21 | **6.31** | **8.18** | 5.56 | 6.62 | **8.20** |
| **DeCo (Ours)** | **6.25** | **5.77** | **7.14** | **6.33** | 6.08 | 7.54 | **7.42** | 6.25 | 7.96 | **7.81** | **6.70** | 8.15 |

## 4.4 ANALYSIS

**Latency and throughput analysis.** To evaluate the efficiency of DeCo, we compare its latency and throughput with several baselines, **including DoLa, OPERA, and VCD based on Greedy, Beam Search, and Nucleus Sampling, respectively**. Figure 6 illustrates the results of this comparison. The findings indicate that DeCo operates within an acceptable efficiency cost, striking a balance between effectiveness and computational overhead. Compared to the basic decoding process,

the latency increase introduced by our method is approximately 1.2 times. In contrast, the latency increases for VCD and OPERA are 1.8 and 5.1 times, respectively. While both VCD and OPERA demonstrate comparable efficacy in mitigating hallucinations, their computational overheads remain relatively high. This highlights the practical value of DeCo, as it can be integrated into real-world applications without significantly compromising efficiency.

**Perturbation in the selected preceding-layer.** To evaluate the effectiveness of the dynamic layer selection method, we introduce a random perturbation strategy. Specifically, for the predetermined preceding layers, we add random values ranging from -5 to 5 to modify the selection of layers. We randomly select 200 images from the MSCOCO dataset and prompt MLLMs to generate descriptions. The results after incorporating the perturbations are presented in Table 5. Notably, the perturbed results demonstrate a significant degradation in performance, further validating the effectiveness of our proposed method.

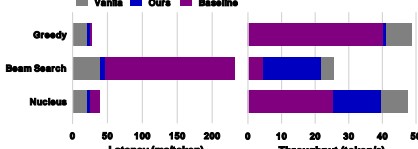

Figure 6: Comparison of latency and throughput across different baselines.

Table 5: Comparison of results between DeCo and perturbed DeCo in image captioning tasks

| Method | InstructBLIP | | MiniGPT-4 | | LLaVA-1.5 | | Qwen-VL | |
|---|---|---|---|---|---|---|---|---|
| | CHAIR$_S$ ↓ | CHAIR$_I$ ↓ | CHAIR$_S$ ↓ | CHAIR$_I$ ↓ | CHAIR$_S$ ↓ | CHAIR$_I$ ↓ | CHAIR$_S$ ↓ | CHAIR$_I$ ↓ |
| DeCo | 39.3 | 12.6 | 32.4 | 9.6 | 38.8 | 11.1 | 44.5 | 11.1 |
| DeCo + ε | 45.6 ↑**6.3** | 14.3 ↑**1.7** | 33.3 ↑**0.9** | 10.1 ↑**0.5** | 42.2 ↑**2.4** | 11.3 ↑**0.2** | 47.0 ↑**2.5** | 12.8 ↑**1.7** |

**Hyperparameter analysis.** Our method incorporates two primary hyperparameters: $\alpha$ and the selection of interval layers. In the experiments, we employ DeCo based on greedy decoding. On the one hand, the hyperparameter $\alpha$ regulates the intensity of early information enhancement. Figure 7(a) illustrates the performance across various $\alpha$ values. We observe that hallucination suppression is most effective when $\alpha$ approximates 0.6. As $\alpha$ increases, the efficacy of DeCo in mitigating hallucinations improves. However, it is crucial to note that excessively high $\alpha$ values may lead to the generation of atypical image descriptions, characterized by repetitive word usage. Notably, we can adjust the value of alpha appropriately to balance the truthfulness and semantic coherence of the responses (e.g., by using lower alpha). Additionally, our approach and the hyperparameter for repetition penalty are orthogonal, which implies that we can introduce the repetition penalty term to mitigate repetition. On the other hand, the layer interval hyperparameter $[a, b]$ determines the candidate layers for inclusion in the enhancement process. We conduct experiments using intervals of four layers, with results presented in Figure 7(b). Our analysis reveals that hallucination suppression for MLLM is negligible in layers 1-16, while layers 20-28 demonstrate substantial mitigation of hallucinations. Notably, layers 29-32 exhibit minimal hallucination suppression, aligning with our findings discussed in Section 2.2. For other families of MLLMs and larger scale MLLMs, the selection of interval layer should be appropriately adjusted based on empirical experimentation.

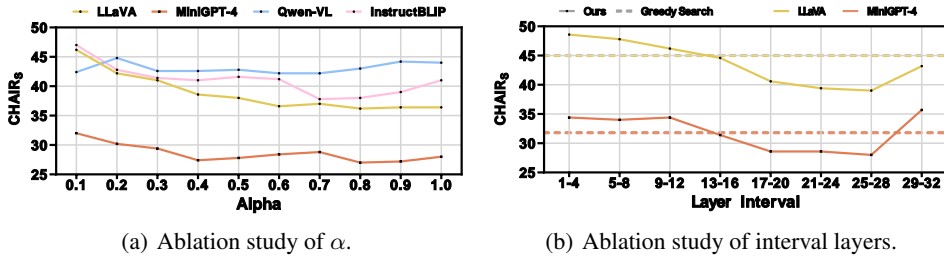

(a) Ablation study of $\alpha$.     (b) Ablation study of interval layers.

Figure 7: Ablation experiment results for hyperparameter $\alpha$ and different interval layers.

**Mitigating snowballing hallucinations.** Snowballing hallucinations are a prevalent issue in the responses generated by MLLMs. This phenomenon occurs when an initial hallucination triggers a

sequence of subsequent errors, leading to a compounding effect that significantly degrades the quality and coherence of the generated text. Figure 8 illustrates a typical example of snowballing hallucinations, where an initial misinterpretation of the visual input propagates through the decoding process, resulting in a highly inconsistent and erroneous output. Our approach can reduce the accumulation of errors and improves the overall consistency and accuracy of the generated responses. The effectiveness of DeCo is further demonstrated through additional cases based on diverse MLLMs, which can be found in Figures 9, 10, 11, and 12 in Appendix G.

## 5    RELATED WORK

### 5.1    MLLM HALLUCINATION MECHANISM

Hallucination in MLLMs, characterized by contradictions between image input and textual output, has been a prevalent issue (Liu et al., 2024a; Chen et al., 2024g). Current research on the mechanism of hallucination in MLLMs focuses on two key aspects: the interaction between images and text at different layers, and the prior bias of the LLM during decoding. Several studies have investigated the role of image-text interaction at different layers in MLLMs. Grad-CAM (Zhang et al., 2024a) visualizations reveal that image-text interaction exists in the preceding layers (1-11) but not in the deep layers. OPERA (Huang et al., 2024) further proposes that the "Aggregation Pattern" leads to hallucination, where visual information from preceding layers is gradually aggregated to anchor tokens, and focusing solely on these tokens during prediction while ignoring visual information leads to a high probability of hallucination in the generated sequence. However, other studies have revealed that MLLMs exhibit biases towards LLM priors, even in the presence of noisy or absent visual information. VCD (Leng et al., 2023) discovers that MLLMs generate high-confidence answers even when the image is noisy or absent, indicating a bias towards LLM priors. Similarly, PAI (Liu et al., 2024b) describes this phenomenon as "Text Inertia" and posits that it stems from existing paradigms that map visual representations onto the text representations as tokens. This leads to an inference process that fails to adequately account for image tokens, resulting in hallucinations.

### 5.2    HALLUCINATION MITIGATION FOR MLLMS

One straightforward approach to mitigate hallucination is to reduce the knowledge gaps and data bias between vision and language during model training. Finetuning-based methods have been explored, focusing on crafting specific datasets (You et al., 2024; Gunjal et al., 2024; Chen et al., 2024f) and alignment training (Sun et al., 2023; Yu et al., 2023; Chen et al., 2023; Li et al., 2023c) to achieve better knowledge alignment between images and text. While these methods have shown promising results, they often require expensive annotated paired data and substantial computational resources.

Hallucination can also be mitigated by post-processing methods, which usually involve using additional tools or self-reflection strategies to revise the response. For instance, LURE (Zhou et al., 2024) detects hallucinations using manually-crafted features and revises the generated text accordingly. Woodpecker (Yin et al., 2023) combines MLLM outputs with an expert VQA model to post-edit hallucinations. However, these methods incur additional inference costs and delays, and require task-specific procedures and prompts to be designed (Xu et al., 2024). Training-free decoding methods have been explored to mitigate hallucination. OPERA (Huang et al., 2024) identifies an abnormal attention pattern that often accompanies hallucinated descriptions and proposes the mitigation method based on this pattern. VCD (Leng et al., 2023) introduces the notion that visual uncertainty increases hallucination and proposes a contrast decoding method to alleviate the issue. VDD (Zhang et al., 2024b) proposes a "Post-Hoc debias" approach that ensures uniform scores for each answer in the absence of an image to mitigate the influence of LLM priors.

## 6    CONCLUSION

In this paper, we demonstrate that MLLMs exhibit an awareness of hallucinated objects, with earlier layers showing higher confidence, while tokens shaped by prior knowledge diminish the likelihood of true tokens in the final layers. Based on this insight, we introduce DeCo, dynamic correction decoding with preceding-layer knowledge to mitigate hallucinations. Extensive experiments demonstrate the efficacy of our approach, which also shows advantages in latency and throughput.

## ACKNOWLEDGMENTS

This work was supported by the National Natural Science Foundation of China (No. 62206246, No. NSFCU23B2055, No. NSFCU19B2027), the Fundamental Research Funds for the Central Universities (226-2023-00138), Yongjiang Talent Introduction Programme (2021A-156-G), CIPSC-SMP-Zhipu Large Model Cross-Disciplinary Fund, Ningbo Science and Technology Special Projects under Grant No. 2023Z212, Information Technology Center and State Key Lab of CAD&CG, Zhejiang University, NUS-NCS Joint Laboratory (A-0008542-00-00), the Scientific Research Starting Foundation of Nanjing University of Aeronautics and Astronautics (No.1015-YAH24096), the High Performance Computing Platform of Nanjing University of Aeronautics and Astronautics, the Ministry of Education, Singapore, under the Academic Research Fund Tier 1 (FY2023) (Grant A-8001996-00-00), NUS-NCS Joint Laboratory (A-0008542-01-00). We gratefully acknowledge the support of Zhejiang University Education Foundation Qizhen Scholar Foundation.

## REPRODUCIBILITY STATEMENT

We have submitted the relevant code in the supplementary materials. The names of the experimental benchmarks, the prompt templates used, and the model's hyperparameter settings can all be found in Section 4. The Appendix C.1 and C.2 provides a detailed description of the experimental setup for the mechanism experiments.

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

## A  LIMITATIONS

**Lack of generalized research.**    Due to the GPU cost consideration, we conduct experiments solely on limited MLLMs, without exploring additional MLLMs or those with larger parameter sizes.

**No free lunch.**    The results shown in Table 4 indicate that our method has a little negative impact on the level of detailedness metric. In future work, we aim to integrate DeCo with other strategies and explore approaches that can effectively balance truthfulness and diversity.

## B  COMPARATIVE ANALYSIS, SUMMARY, AND FUTURE DIRECTIONS

Here, we compare our work with previous works, summarize and speculate on the underlying causes of hallucinations in LLMs and MLLMs, and provide insights into future directions.

**Comparison of previous mechanism findings.**    Existing studies suggest that MLLMs may focus more on visual tokens in the early layers while paying greater attention to textual tokens in the later layers (Zhang et al., 2024a; Chen et al., 2024c). The aggregation pattern is typically positively correlated with hallucinations and tends to emerge at deeper layers (Huang et al., 2024). These conclusions align with our findings, suggesting that MLLMs exhibit a better ability to perceive visual information in the preceding-layers compared to the final layers. In the detection of hallucinations in LLMs, some studies employing probing techniques have found that the intermediate layers exhibit the best detection performance (Chen et al., 2024a; Orgad et al., 2024; Chen et al., 2024e; Lu et al., 2024b), a finding similar to our Finding 1. This suggests that the hallucination mechanisms in LLMs and MLLMs may share underlying similarities.

**Comparison of previous hallucination mitigation methods.**    Our work shares a similar assumption with OPERA (Huang et al., 2024) and VCD (Leng et al., 2023), positing that the knowledge priors inherent in MLLMs may suppress the model's ability to comprehend visual information. However, our approach is comparatively simpler than that of OPERA (Huang et al., 2024) and VCD (Leng et al., 2023). Additionally, our work differs from the assumption in unimodal LLMs, where the semantic information present in the shallow layers interferes with factual recall in the final layer (Chuang et al., 2024; Chen et al., 2024d). However, our method is actually parallel to previous approaches and can be combined to achieve better results.

**Summary.**    Combining current research, we speculate that this phenomenon observed in both LLMs and MLLMs may be due to characteristics of the Transformer architecture, specifically the anchor token effect in the attention mechanism (Huang et al., 2024; Wang et al., 2023b), which leads to information loss when processing long sequences. For instance, in MLLMs, a single token may be insufficient to summarize information from extended sequences of visual tokens. Another work suggest that the knowledge overshadowing of multiple conditions within the query leads to hallucinations in LLM (Zhang et al., 2024c). In multimodal settings, image information represents a distinct condition. When the textual modality overshadows the condition related to the image, it can result in hallucinations in visual perception. Essentially, this reflects a loss of information flow within the attention mechanism. Overall, from an architectural perspective, hallucinations in both LLMs and MLLMs arise due to the imperfect of handling such interactions within the attention patterns of the Transformer.

**Future directions.**    Overall, the present works and ours work reveals notable similarities in the internal patterns of hallucination between LLMs and MLLMs. In future research, we will adopt a unified perspective to investigate the underlying causes of hallucinations in both LLMs and MLLMs.

## C    DETAILED EXPERIMENTAL SETUP

### C.1    DETAILED SETTINGS FOR FINDINGS 1

In the probing experiment, we utilize the pipeline proposed in the POPE (Li et al., 2023b) to construct 1,200 balanced positive and negative sample pairs from the MSCOCO dataset as training data for the probe classifier, where each sample consists of an object accompanied by a label indicating its existence or non-existence. (**Note**: There is no overlap between the training data and the evaluation data for object hallucination proposed by the POPE). We select the AMBER dataset (Wang et al., 2023a), which has a different distribution from the MSCOCO dataset, to test whether our conclusions can generalize. The AMBER dataset contains 1,004 carefully annotated images, each labeled with existent objects as well as non-existent objects. We use the prompt "`Describe the image.`" to generate raw responses from LLaVA-1.5 on the images and then extract all object category tokens and label them with whether they exist. Given that the training set contains only 80 object categories, we denote the object tokens in test data belonging to these 80 categories as in-distribution (in-dist), while the remaining tokens are categorized as out-of-distribution (OOD).

Previous work (Karamcheti et al., 2024) has demonstrated that increasing the resolution of the vision encoder enhances the visual comprehension capabilities of MLLMs. In our study, we compare LLaVA trained with a resolution of 224px against the original LLaVA with a resolution of 336px in probing experiments. Notably, the language model's weights differ between the two MLLMs, although both initial models are based on Vicuna-1.5-7b. Our results, as illustrated in the Figure 1(b), further affirm the scaling law associated with visual resolution, while also providing indirect validation of the reliability of the probing experiments.

### C.2    DETAILED SETTINGS FOR FINDINGS 2

In the early exit experiment, we randomly select 500 images from MSOCO and use random prompts (shown in Table 6) to elicit raw responses from LLaVA-1.5-7b. We then extract all non-existent objects along with their corresponding preceding text. Specifically, for the sentence "`Additionally, there is a car.`", we extract the hallucinated object token "`car`" and the preceding text "`Additionally, there is a`". We re-input the preceding text into the MLLM and observe the changes in its internal state when predicting the next token. We denote that a total of $K$ preceding texts are selected, with the j-th preceding text denoted as $s^j$.

Table 6: Randomly prompts.

| Prompts |
| --- |
| Describe the image. |
| Please describe this image in detail. |
| Generate a caption for this image. |

# D    EVALUATION RESULTS IN AMBER

The AMBER image caption dataset consists of 1,004 images, each accompanied by meticulously annotated labels. These annotations include all objects present in the images, as well as some potential hallucinated objects. AMBER employs four evaluation metrics: CHAIR (the proportion of generated hallucinated objects among all objects), Cover (the coverage of generated objects against all ground truth objects), Hal (the proportion of hallucinations among all generated captions), and Cog (the overlap ratio with potential hallucinated objects). Lower values of CHAIR, Hal, and Cog indicate higher truthfulness for the MLLMs, while a higher Cover value signifies better diversity. We compare Deco with the baselines on the LLaVA-1.5-7b. The results are as shown in Table 7. The results reveal that Deco demonstrates a significant advantage in truthfulness, although its diversity is somewhat lacking, yet remains within an acceptable range.

Table 7: Results of using DeCo on the AMBER image caption dataset with LLaVA-1.5-7b.

| Decoding | Method | LLaVA-1.5 | | | |
|---|---|---|---|---|---|
| | | CHAIR ↓ | Cover ↑ | Hal ↓ | Cog ↓ |
| Greedy | Vanilla | 8.2 | 48.9 | 34.3 | 4.0 |
| | DoLa | 8.0 | **50.8** | 37.5 | 4.3 |
| | **DeCo (Ours)** | **6.6** ↓1.6 | 47.5 ↓1.4 | **28.1** ↓6.2 | **2.8** ↓1.2 |
| Beam Search | Vanilla | 7.1 | **50.7** | 32.4 | 3.8 |
| | OPERA | 6.4 | 49.0 | 27.5 | 2.9 |
| | **DeCo (Ours)** | **6.3** ↓0.8 | 46.8 ↓3.9 | **25.1** ↓7.3 | **2.4** ↓1.4 |
| Nucleus | Vanilla | 10.2 | 50.2 | 43.3 | 4.5 |
| | VCD | 9.0 | **51.7** | 40.2 | 4.4 |
| | **DeCo (Ours)** | **8.3** ↓1.9 | 48.0 ↓2.2 | **37.5** ↓5.8 | **3.4** ↓1.1 |

# E    ABLATION RESULTS OF DYNAMIC SOFT MODULATION

To quantify the effect of soft modulation, we remove the "`max_prob`" term and use greedy decoding to describe the images. The images and prompts used in the ablation experiment are consistent with the setup in Table 2, and the ablation results are presented in the Table 8. Additionally, we provide illustrative cases that demonstrate how soft modulation helps prevent abrupt changes in logits, as shown in the Figure 13.

Table 8: Ablation study of dynamic soft modulation.

| Method | LLaVA-1.5 | | Qwen-VL | |
|---|---|---|---|---|
| | CHAIR$_S$ ↓ | CHAIR$_I$ ↓ | CHAIR$_S$ ↓ | CHAIR$_I$ ↓ |
| DeCo | 37.8 | 11.1 | 42.2 | 10.7 |
| DeCo (wo/max_probs) | 41.2 ↑3.4 | 11.6 ↑0.5 | 45.8 ↑3.6 | 12.3 ↑1.6 |

# F    RESULTS OF MMVET

We evaluate the performance of LLava-1.5-7B and Qwen-VL-7B on a comprehensive benchmark MMVet and use their default settings based on nucleus sampling. The results are as shown in Table 9.

Table 9: Results of using DeCo on MMVet with LLaVA-1.5-7b and Qwen-VL-7b.

| Model | Method | Rec ↑ | OCR ↑ | Know ↑ | Gen ↑ | Spat ↑ | Math ↑ | Total ↑ |
|---|---|---|---|---|---|---|---|---|
| LLaVA-1.5 | Vanilla | 28.8 | 14.1 | 15.5 | 16.4 | 15.6 | 3.5 | 23.6 |
| | **DeCo (Ours)** | **32.1** ↑3.3 | **21.5** ↑7.4 | **18.6** ↑3.1 | **20.7** ↑4.3 | **23.7** ↑8.1 | **11.2** ↑7.7 | **27.9** ↑4.3 |
| Qwen-VL | Vanilla | **51.8** | 35.3 | **41.0** | 35.6 | 38.1 | 19.2 | 45.7 |
| | **DeCo (Ours)** | 50.5 ↓1.3 | **38.2** ↑2.9 | 38.8 ↓2.2 | 33.8 ↓1.8 | **41.7** ↑3.6 | **26.5** ↑7.3 | **46.3** ↑0.6 |

# G    CASE ANALYSIS ACROSS DIVERSE MLLMS

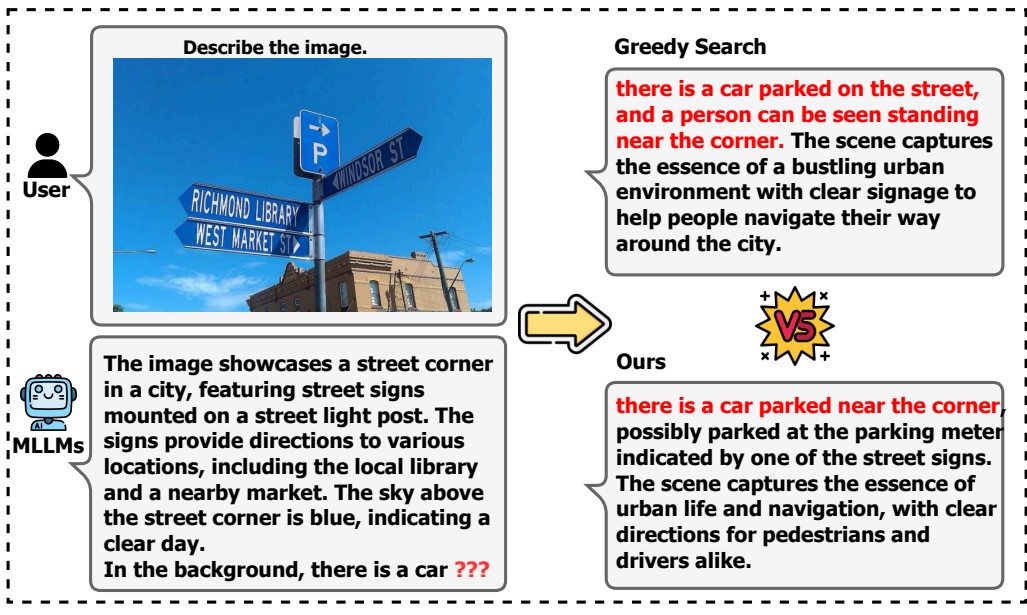

Figure 8: The case of mitigating snowballing hallucination with DeCo.

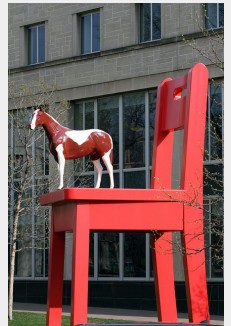

Instruction: Please describe this image in detail.

InstructBlip (Greedy):
The image features a large red chair with the shape of an animal, specifically in this case being represented by two horses. One horse is standing on top and another one appears to be sitting atop it from behind its back legs towards front side viewing angle. There are several people visible throughout different parts around or near these animals: three individuals can been seen walking along either sides close together; there's also someone further away who seems more distant but still within reachable distance for interaction if needed. Additionally, some cars parked nearby add movement into scene as they come closer while driving past each other during their daily activities outside.

InstructBlip (Ours):
The image features a large red chair with a horse statue placed on top of it. The horse statue is positioned on the chair's seat, making it appear as if the horse is actually sitting on the chair. The chair is placed on a grassy area, likely outside, and there are several trees visible in the background. The scene is set up as a sculpture, showcasing a unique and creative way of displaying a horse statue.

**Input Image**

---

Instruction: Please describe this image in detail.

InstructBlip (Beam search):
The image depicts a group of three motorcyclists riding down a winding road through a mountainous area. The road is surrounded by tall, rocky cliffs on both sides, creating a dramatic and scenic setting. The motorcyclists are wearing helmets and appear to be enjoying the ride, taking in the breathtaking views around them. There are two motorcycles visible in the scene, with one closer to the center of the image and the other slightly off to the right side. In addition to the motorcycles, there are several cars parked along the side of the road, providing additional transportation options for those exploring the area.

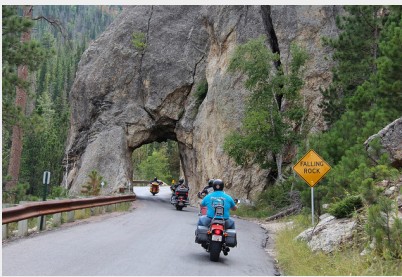

**Input Image**

InstructBlip (Ours):
The image depicts a group of motorcycles traveling down a roadway, passing through a tunnel. The motorcycles are ridden by three people who are enjoying the scenic route through the tunnel. The motorcycles are positioned in various positions along the roadway, with one motorcycle being closer to the center of the image, another motorcycle slightly behind it, and another motorcycle further away from the center of the image. The third motorcycle is positioned further away from the others, but still visible in the background. The roadway itself is surrounded by trees and rocks, adding to the scenic nature of the scene.

---

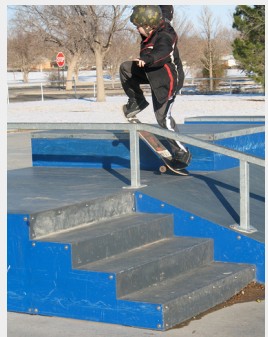

Instruction: Please describe this image in detail.

InstructBlip (Nucleus):
The image captures a skateboarder skillfully riding a rail in a blue skate park, likely practicing his tricks. The skateboarder is wearing a helmet, providing safety for the activity. There are multiple cars present in the background, including two parked on either side of the skate park. Additionally, there are two benches scattered around the scene, likely providing seating for the skateboarder and other park visitors.

InstructBlip (Ours):
The image depicts a skateboarder skillfully skating on a rail near a blue railing at a skateboarding park. The skateboarder is wearing a helmet and appears to be engaging in a trick, performing a rail grind on the rail. The skateboarding park is well-equipped with several skateboarding obstacles, including a skateboarding ramp, a skateboarding rail, and a skateboarding ledge. These obstacles are designed to challenge skateboarders and provide them with the opportunity to practice their skills and showcase their skills in various ways.

**Input Image**

Figure 9: DeCo's performance in reducing hallucinations of InstructBlip-7B on three basic decoing methods.

Instruction: Please describe this image in detail.

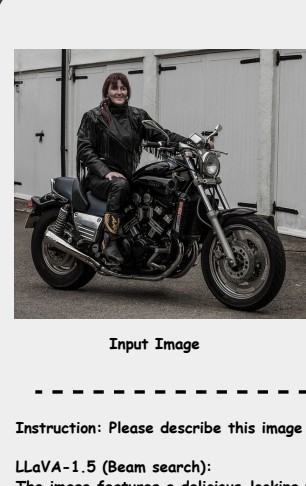

**Input Image**

LLaVA-1.5 (Greedy):
The image features a woman sitting on a motorcycle, which is parked in front of a white building. She is wearing a leather jacket and appears to be posing for the camera. The motorcycle is positioned in the foreground, with the woman sitting comfortably on the seat.

In the background, there is a car parked further away from the motorcycle. The scene captures the woman's enthusiasm for her motorcycle and her outdoor adventures.

LLaVA-1.5 (Ours):
The image features a woman sitting on a motorcycle parked outside a building. She is positioned in front of the motorcycle, occupying a majority of the frame. The motorcycle itself occupies a large portion of the scene, extending from left to right.

The woman appears to be dressed in motorcycle gear, possibly preparing for a ride or already having completed one. The motorcycle's positioning suggests that it might be parked outside a garage or other similar location where motorcycle enthusiasts gather.

- - - - - - - - - - - - - - - - - - - - - - - - - - - - - - - - - - - - - - - - -

Instruction: Please describe this image in detail.

LLaVA-1.5 (Beam search):
The image features a delicious-looking pizza sitting in a box on a dining table. The pizza is topped with tomatoes and cheese, making it an appetizing sight. Next to the pizza, there is a bottle of beer, which complements the meal perfectly.
In addition to the pizza and beer, there is a knife placed on the table, likely for cutting and serving the pizza slices. The dining table occupies most of the space in the image, emphasizing the focus on the pizza and beer.

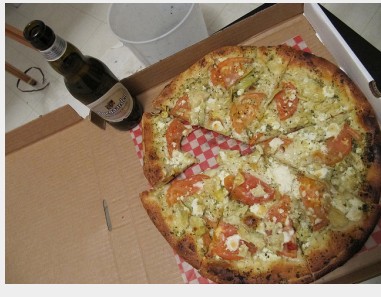

**Input Image**

LLaVA-1.5 (Ours):
The image features a delicious looking pizza sitting inside a box, ready to be eaten. The pizza is topped with tomatoes and cheese, making it look delicious and freshly prepared. Next to the pizza box, there is a bottle of beer, adding to the overall enjoyment of the meal. The scene appears to be ready for someone to enjoy a delicious combination of pizza and beer.

- - - - - - - - - - - - - - - - - - - - - - - - - - - - - - - - - - - - - - - -

Instruction: Please describe this image in detail.

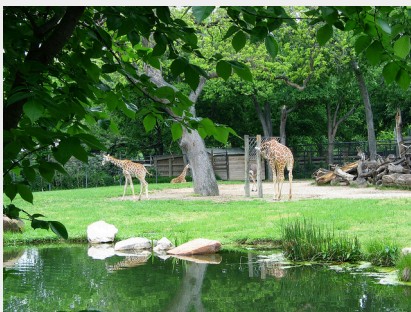

**Input Image**

LLaVA-1.5 (Nucleus):
The image features a group of four giraffes in a lush green field, surrounded by trees and a pond. The giraffes are standing close to each other, with one on the left side, two in the middle, and one on the right side of the scene. They appear to be enjoying their time in the open field, possibly in a zoo or wildlife sanctuary.

In addition to the giraffes, there are two birds visible in the scene, one near the left side and the other closer to the center. The birds are likely part of the wildlife in the area. The overall atmosphere of the image is serene and peaceful, showcasing the beauty of the natural environment.

LLaVA-1.5 (Ours):
The image depicts a group of four giraffes in a zoo enclosure, enjoying their time together in the grassy area. They are spread out throughout the scene, with one giraffe near the left side of the enclosure, another on the right side, one close to the middle, and another giraffe further back in the background.

The zoo enclosure features a pond in the middle, where the giraffes can drink water or cool off during hot days. There are also several trees within the enclosure, providing the animals with shade and a natural habitat feel.

Figure 10: DeCo's performance in reducing hallucinations of LLaVA-1.5-7B on three basic decoing methods.

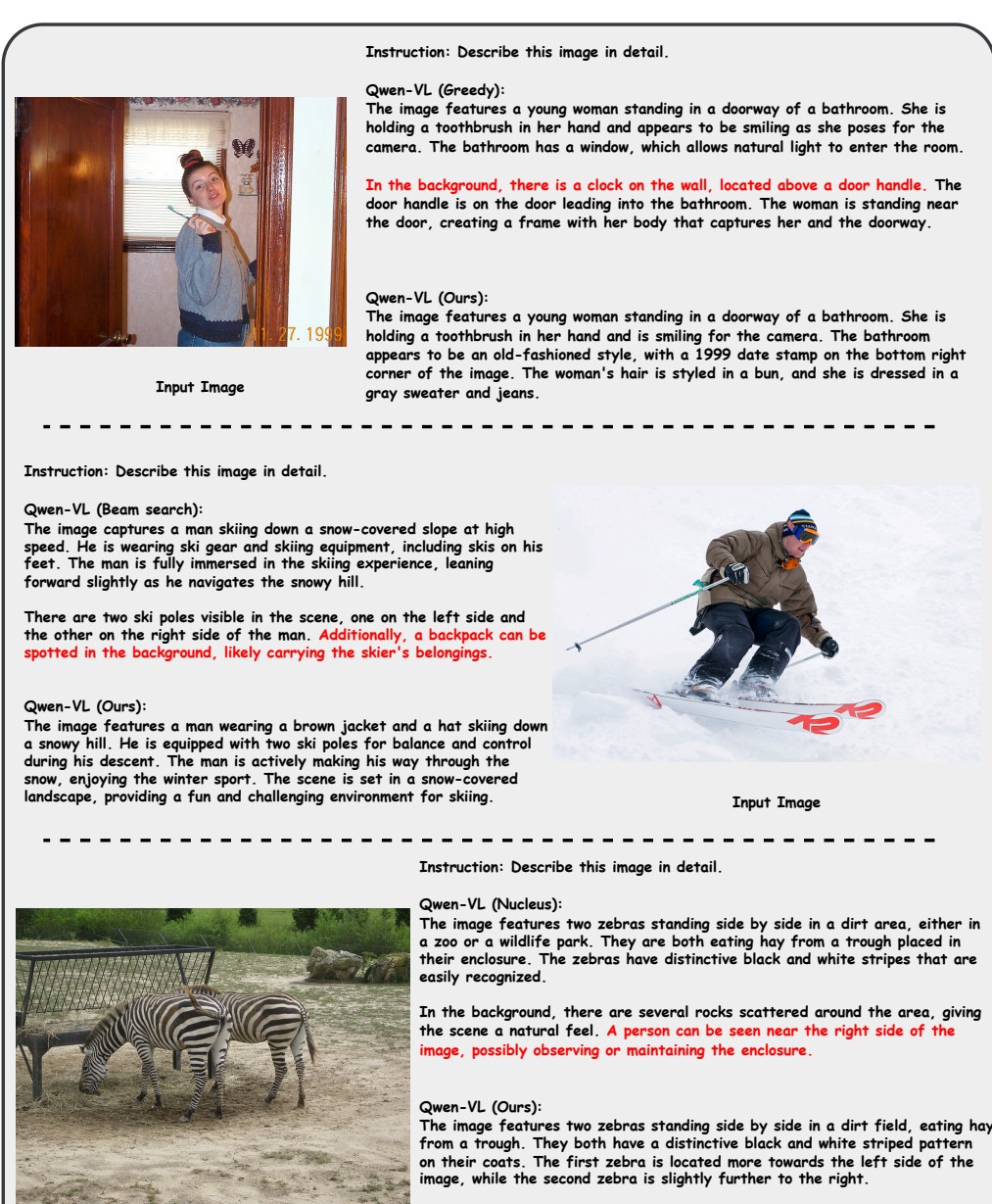

Figure 11: DeCo's performance in reducing hallucinations of Qwen-VL-7B on three basic decoing methods.

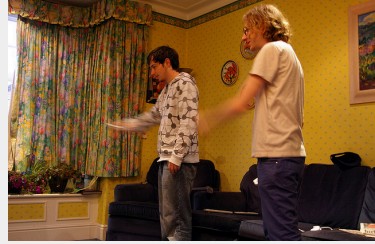

Instruction: Please describe this image in detail.

MiniGPT4 (Greedy):
This image shows two men standing in a living room. One man is holding a white object in his right hand, while the other man is standing behind him with his arms crossed in front of his chest. The room is decorated with yellow curtains and a blue couch. The furniture is cluttered with various objects, including a lamp, a vase, and a book. The walls are adorned with yellow and white wallpaper. The overall atmosphere of the image is casual and relaxed.

MiniGPT4 (Ours):
This image depicts two men standing in a room with yellow walls and curtains. One man is holding a white object in his right hand while the other man is standing behind him with his arms crossed in front of him. The furniture in the room includes a blue couch and a chair. The room appears to be well lit with sunlight coming through the window.

**Input Image**

- - - - - - - - - - - - - - - - - - - - - - - - - - - - - - - - - - - - - - - - - - - - - - - -

Instruction: Please describe this image in detail.

MiniGPT4 (Beam search):
The image shows a large clock tower in the middle of a city street. The clock tower is made of metal and has two large clock faces on it. The clock tower is surrounded by tall buildings on either side of it. There is a black car parked on the sidewalk next to the clock tower. The car is facing the clock tower and has its lights on. There are people walking on the sidewalk in front of the clock tower. Some of them are looking up at the clock tower while others are walking past it. The buildings on either side of the clock tower have large windows and balconies. The street is lined with trees and streetlights.

MiniGPT4 (Ours):
This image depicts a large clock tower in the middle of a city street. The clock tower is made of metal and has two clock faces on it. The clock tower is surrounded by buildings and cars passing by on the street below it. There is a black car parked on the sidewalk next to the clock tower.

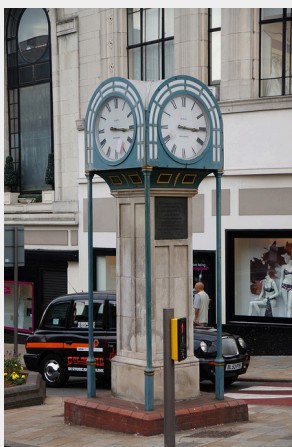

**Input Image**

- - - - - - - - - - - - - - - - - - - - - - - - - - - - - - - - - - - - - - - - - - - - - - - -

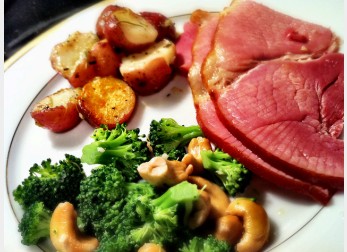

Instruction: Please describe this image in detail.

MiniGPT4 (Nucleus):
The image shows a white plate with a piece of meat, a bunch of broccoli, and some cashews. There is a knife and fork on the side of the plate, and a glass of water on the other side. The plate is on a brown tablecloth and the background is a brown wall.

MiniGPT4 (Ours):
The image shows a plate of food consisting of broccoli, potatoes, and meat. The plate is white with a black rim and appears to be on a table or countertop. The food is arranged in rows with broccoli on one side of the plate and potatoes on the other side. A piece of meat is in the center of the plate.

**Input Image**

Figure 12: DeCo's performance in reducing hallucinations of MiniGPT4-7B on three basic decoing methods.

**Instruction: Please describe this image in detail.**

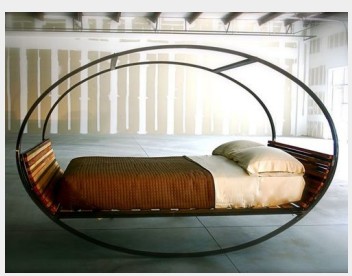

**Ours:**
The image depicts a bedroom scene where a bed occupies a large portion of the space. The bed itself is unique, resembling a circular shape with a bedroom divider in its center. The bedroom divider separates the bed from other parts of the room.

There are two pillows on the bed, one on each side, adding comfort to the bedding arrangement. The bedroom also features a wall with a few windows, allowing natural light to illuminate the room.

**Ours (wo/max_probs):**
The image depicts a bedroom scene where a bed occupies most of the space. The bed is unique due to its shape and style; it resembles a circular bed frame suspended above the floor. It appears to be made of metal and has a bedding arrangement consisting of pillows and blankets placed on top of it.

There are two pillows visible on the bed, one located near the middle of the bed frame and another one towards the bottom part of it. Additionally, two blankets can be seen on the bed frame, one occupying the middle part of it and another one slightly higher than the first one.

**Input Image**

---

**Instruction: Please describe this image in detail.**

**Ours:**
The image depicts two giraffes standing together inside a zoo enclosure. They are both tall and majestic animals, one occupying a majority of the space in the image while the other occupies a smaller portion. Both giraffes are feeding from a feeding basket hung high above them, likely providing them with food to eat.

The zoo enclosure features concrete walls and a concrete floor, creating a solid structure for the animals to stand on. The scene captures the beauty of these two majestic animals interacting with each other and feeding together.

**Ours (wo/max_probs):**
The image depicts two giraffes standing together inside a zoo enclosure. They are both tall and majestic animals, one occupying the left side of the frame while the other occupies the right side. Both giraffes are feeding on hay placed high above them, likely hung from a feeding pole or similar device.

The zoo enclosure features concrete walls and a concrete floor, creating a solid structure for the animals to stand within. Additionally, there are two clocks visible in the background, one located above the other, likely meant for zoo staff to track time during feeding or other activities within the enclosure.

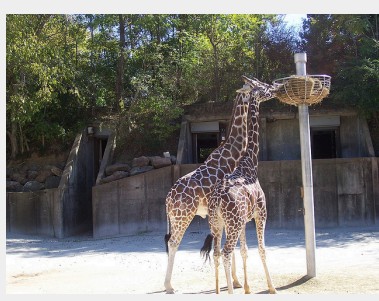

**Input Image**

---

**Instruction: Please describe this image in detail.**

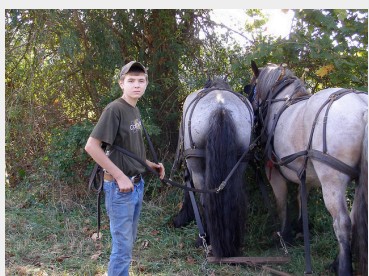

**Ours:**
The image depicts a young boy standing next to two horses in a grassy field. The boy is holding the horses' reins, guiding them as they stand together. The horses are positioned close to each other, one occupying the left side of the field and the other occupying the right side.

The boy is wearing jeans and a green shirt, and he appears to be enjoying his time with the horses. The scene captures a moment of bonding between the boy and the horses, showcasing the connection between humans and animals.

**Ours (wo/max_probs):**
The image depicts a young boy standing next to two horses, one white and one black, both equipped with harnesses. The boy seems to be handling both horses simultaneously, possibly preparing them for a ride or simply admiring them.

The boy is positioned between the two horses, one occupying the left side of the frame while the other occupies the right side of the frame. Both horses are equipped with harnesses, one harnessed nearer to the boy's left side while the other harnessed nearer to his right side.

**Input Image**

Figure 13: Analysis of cases of soft modulation. Soft modulation avoids the hallucination phenomena and erroneous semantics caused by abrupt changes in logits.

Table 10: The prompt used for GPT-4o evaluation adopted from Leng et al. (2023); Huang et al. (2024); Liu et al. (2024b)

---

**GPT-4o Prompt**

You are required to score the performance of two AI assistants in describing a given image. You should pay extra attention to the hallucination, which refers to the part of descriptions that are inconsistent with the image content, such as claiming the existence of something not present in the image or describing incorrectly in terms of the counts, positions, or colors of objects in the image. Please rate the responses of the assistants on a scale of 1 to 10, where a higher score indicates better performance, according to the following criteria:

1: Accuracy: whether the response is accurate with respect to the image content. Responses with fewer hallucinations should be given higher scores.

2: Detailedness: whether the response is rich in necessary details. Note that hallucinated descriptions should not count as necessary details.

Please output the scores for each criterion, containing only two values indicating the scores for Assistant 1 and 2, respectively. The two scores are separated by a space. Following the scores, please provide an explanation of your evaluation, avoiding any potential bias and ensuring that the order in which the responses were presented does not affect your judgment.

[Assistant 1]
{Response of Assistant 1}
[End of Assistant 1]

[Assistant 2]
{Response of Assistant 2}
[End of Assistant 2]

Output format:
Accuracy: <Scores of the two answers>
Reason:

Detailedness: <Scores of the two answers>
Reason:

---

Table 11: The prompt used for GPT-4o to evaluate coherence.

---

**GPT-4o Prompt**

You are required to score the coherence of two AI assistants in describing a given image. Please rate the responses of the assistants on a scale of 1 to 10, where a higher score indicates better coherence.

[Assistant 1]
{Response of Assistant 1}
[End of Assistant 1]

[Assistant 2]
{Response of Assistant 2}
[End of Assistant 2]

Output format: Coherence: <Scores of the two answers >
Reason:

---

