# OpenReview forum: "MLLM can see? Dynamic Correction Decoding for Hallucination Mitigation"
_ICLR.cc/2025/Conference — ICLR 2025 Poster_

### Official Review · Reviewer_QBQj · 2024-10-18

**Soundness:** 3
**Presentation:** 3
**Contribution:** 3
**Rating:** 6
**Confidence:** 4

**Summary:**

The paper investigates why MLLMs generate hallucinations, particularly in image captioning tasks, and introduces DeCo, an innovative method that leverages knowledge from earlier layers to reduce hallucinations during inference. However, I still have some concerns about this article, specifically in regard to the weaknesses.
If these concerns are addressed, I will consider raising my score.

**Strengths:**

- The paper provides a detailed examination of why MLLMs generate non-existent objects, offering valuable insights into the hallucination issue in image captioning tasks.

-  The introduction of DeCo is innovative, using preceding-layer knowledge to reduce hallucinations during inference, effectively improving output accuracy.

- The method of probing across transformer layers reveals how hallucinations emerge in later layers, helping to understand MLLM behavior better.

**Weaknesses:**

- I am confused by the experimental results about POPE in Table 3, as they do not seem to fully align with the result from LLAVA 1.5.
- The authors did not perform more extensive evaluations on comprehensive benchmark such as MMbench and MMVet, which are crucial for assessing the model's overall performance.

**Questions:**

- Could you clarify whether Finding 1 in Section 2.1 in the section is related to the methodology of the paper? It seems that embedding-level knowledge wasn't used to assist the model.
- Could you follow LLAVA’s setting and conduct more extensive evaluations on comprehensive benchmarks like MMbench and MMVet, given their importance for assessing the model's overall performance?
-  Could the authors clarify the specific settings followed in the experiments presented in Table 3?  How do these settings differ from those used in LLaVA?
- Is this decoding method useful in more advanced VLLMs, such as Qwen-VL, VILA, etc.?

---

### Official Review · Reviewer_jCUa · 2024-11-01

**Soundness:** 2
**Presentation:** 3
**Contribution:** 2
**Rating:** 6
**Confidence:** 4

**Summary:**

This paper makes the observation that MLLMs' internal prior suppresses the visual information, thus leading to hallucination. Besides, they empirically observe that intermediate layers may have less such suppression. Motivated by this observation, this work proposes to combine intermediate logits with final layer projection, and demonstrate improvement in reducing hallucination via empirical study.

**Strengths:**

- This work makes an interesting observation of how visual information exists in intermediate layers, and then overridden by knowledge prior closer to the output
- The proposed mitigation method is lightweight and efficient.
- The experimental results are in general better than baselines.

**Weaknesses:**

Although the presentation has a focus about image-conditioned generative language model, the methodology for finding 1 and 2, as well as the proposed layer selection and probability correction, are modality agnostic. The findings are mostly empirical, and it's unclear whether this is a general phenomenum for other models in the same size, nor for models in other sizes.

There has been quite a few literature in studying LLM's internal presentation and hallucination, only selectively listing a few as [1-5] . What the multi-modal setting brings is the strong conditional dependency, while for text-only use cases there might or might not be informative conditions. An analytical comparison on how an MLLM focuses or ignores input conditions can be more informative and persuasive in supporting the methodology.

In L191-L201 the paper compares the token output with and without the image condition. However this has been studied thouroughly in [6], which also proposes hallucination detection and mitigation method.

The method design also seems ad-hoc, there are thresholds in Eq2 and Eq3, layer interval a, b in Eq4 and the weight $\alpha$ in Eq7. Together they contribute to amplifying the concern in the generalizability of the proposed method.

I suggest to connect the empirical evidences in this paper to 1/ evidences from other papers with the same spirit, and 2/ the unique property and behavior of conditional multi-modal modeling.

**Reference**

[1] Liu, Junteng, et al. "On the Universal Truthfulness Hyperplane Inside LLMs." arXiv preprint arXiv:2407.08582 (2024).

[2] Li, Kenneth, et al. "Inference-time intervention: Eliciting truthful answers from a language model." Advances in Neural Information Processing Systems 36 (2024).

[3] Zhang, Tianhang, et al. "Enhancing uncertainty-based hallucination detection with stronger focus." arXiv preprint arXiv:2311.13230 (2023).

[4] Azaria, Amos, and Tom Mitchell. "The internal state of an LLM knows when it's lying." arXiv preprint arXiv:2304.13734 (2023).

[5] Duan, Hanyu, Yi Yang, and Kar Yan Tam. "Do LLMs Know about Hallucination? An Empirical Investigation of LLM's Hidden States." arXiv preprint arXiv:2402.09733 (2024).

[6] Favero, Alessandro, et al. "Multi-modal hallucination control by visual information grounding." Proceedings of the IEEE/CVF Conference on Computer Vision and Pattern Recognition. 2024.

**Questions:**

- It's unclear to me how is the affine layer $\phi$, in L104, initialized and trained, if at all? If it needs training, then it seems each layer needs such a layer. If it doesn't, then how do we make sure that resentation across layers can share the same mapping to present token probability?
- POPE evaluates answers in Yes/No. How could decoding strategy have impact on the performance for this benchmark?

---

### Official Review · Reviewer_tgHP · 2024-11-03

**Soundness:** 3
**Presentation:** 3
**Contribution:** 3
**Rating:** 6
**Confidence:** 4

**Summary:**

This paper demonstrates that while MLLMs may produce incorrect target outputs in the final layer, they effectively recognize visual objects in the preceding layers. The authors propose a dynamic correction decoding method for MLLMs (DeCo), which adaptively selects relevant preceding layers and proportionally integrates their knowledge into the final layer to adjust the output logits. The proposed method outperforms existing approaches on public datasets.

**Strengths:**

1. The motivation seems interesting.
2. The paper is well written and easy to follow. The diagrams are essential to understanding this paper.
3. This paper achieves good results on existing datasets.
4. The main technical pipeline is clear.

**Weaknesses:**

1. Although the experiments indicate improved performance in preceding layers, I am concerned about the coherence and richness of the text generated at these stages. Could you provide further evaluation metrics for text quality, such as BLEU or other relevant scores?
2. In Figure 1(b), the interval [10, 20] appears optimal, yet in Figure 7(b), [17, 28] shows better performance. Could you clarify this discrepancy?
3. Could you provide more evidence to demonstrate how dynamic soft modulation prevents abrupt changes in logits? Additional ablation studies might further substantiate this claim.
4. Could you share detailed MME results to highlight the method's performance across different subtasks?

**Questions:**

My primary concern lies in the potentially low quality of text generated from the preceding layers. I will be happy to raise my score if my current questions and concerns can be addressed.

---

### Official Review · Reviewer_vNi9 · 2024-11-03

**Soundness:** 3
**Presentation:** 4
**Contribution:** 3
**Rating:** 6
**Confidence:** 5

**Summary:**

The paper introduces DeCo (Dynamic Correction Decoding), a decoding technique to mitigate hallucinations in Multimodal Large Language Models (MLLMs). The authors identify that MLLMs are capable of recognizing objects in earlier layers, but this recognition is suppressed in deeper layers by strong language model priors, which leads to hallucinations. DeCo dynamically integrates the output from preceding layers, which contain higher probabilities for ground-truth tokens, into the final layer logits to enhance visual grounding and suppress hallucinations. Experimental results on datasets such as CHAIR, POPE, MME and GPT-4o assisted evaluation demonstrate DeCo’s significant improvements over baselines in hallucination suppression across multiple MLLMs, with manageable latency increases, highlighting its practical applicability.

**Strengths:**

- The authors demonstrate through probing experiments that MLLMs can recognize objects in earlier layers but tend to “forget” this information due to language model priors in deeper layers, leading to hallucinations. This insight offers a novel layer-wise perspective on the hallucination mechanism in MLLMs.
- The figures illustrating token probabilities across transformer layers effectively highlight the trends for hallucinated versus non-hallucinated tokens, making the analysis accessible and informative.
- Compared to existing methods like VCD and OPERA, DeCo achieves similar or better hallucination suppression with lower latency overhead, enhancing its practicality for real-world applications.
- Evaluation across diverse benchmarks (CHAIR, POPE, and MME) and several models (InstructBLIP, MiniGPT-4, LLaVA-1.5, and Qwen-VL) provides a well-rounded assessment of DeCo’s effectiveness.

**Weaknesses:**

- In Figure 9, the response includes awkward repetition, with "The horse statue is positioned on top of the chair" stated multiple times. This raises questions about the effectiveness of the chosen α\alphaα value in avoiding repetitive language, as the authors indicated that high \alpha values could increase repetition.
- In Figure 10, DeCo reduces a significant hallucination (misidentifying a lift as a "chair"), but the output still contains a hallucination about "several other people visible in the background." This discrepancy between benchmark performance and qualitative examples suggests that DeCo’s effectiveness might not fully translate into consistently accurate real-world responses.
- For each time step tt, language tokens that are not related to the visual input but are essential for sentence generation may be influenced as they pass through the proposed method. There appears to be a lack of investigation into the nature of this influence.

**Questions:**

- Given that DeCo's effectiveness depends on selecting an optimal layer range (e.g., 20-28), does the layer range need tuning for different MLLMs?
- Could you provide more details on the selection process for the 500 images used in experiments? Additionally, which split(s) were used to determine and evaluate the hyperparameters, and were any specific criteria applied for these selections?

---

### Meta-Review · Area_Chair_98HL · 2024-12-20

**Metareview:**

This works shows that though MLLMs could generate incorrect outputs in the final layer, they could effectively recognise objects in preceding layers. Thus, this work introduces a dynamic correction decoding approach for MLLMs to adaptively choose relevant layers and integrate their knowledge into the final layer to adjust outputs. This insight provides a new layer-wise view on the hallucination problem in MLLMs. The proposed method is lightweight and efficient, and the results show the efficacy of such a method. All the reviewers recommend acceptance. In the camera ready version, authors need to carefully improve the paper following reviewers suggestions, including adding in-depth analysis of the model and statistical analysis.

**Additional Comments On Reviewer Discussion:**

Authors replied reviewers' questions about adding implementation details, adding additional metrics, adding additional benchmarks etc. After rebuttal, all reviewers are happy with the work.

---

### Decision · Program_Chairs · 2025-01-22

Accept (Poster)